# Shallow Donor Impurity States with Excitonic Contribution in GaAs/AlGaAs and CdTe/CdSe Truncated Conical Quantum Dots under Applied Magnetic Field

**DOI:** 10.3390/nano11112832

**Published:** 2021-10-25

**Authors:** Lorenz Pulgar-Velásquez, José Sierra-Ortega, Juan A. Vinasco, David Laroze, Adrian Radu, Esin Kasapoglu, Ricardo L. Restrepo, John A. Gil-Corrales, Alvaro L. Morales, Carlos A. Duque

**Affiliations:** 1Grupo de Investigación en Teoría de la Materia Condensada, Universidad del Magdalena, Santa Marta 470004, Colombia; lorenzpulgar@gmail.com (L.P.-V.); jsierraortega@gmail.com (J.S.-O.); 2Instituto de Alta Investigación, CEDENNA, Universidad de Tarapacá, Casilla 7D, Arica 1000000, Chile; juan.vinascos@udea.edu.co (J.A.V.); david.laroze@gmail.com (D.L.); 3Department of Physics, “Politehnica” University of Bucharest, 313 Splaiul Independenţei, RO 060042 Bucharest, Romania; radu@physics.pub.ro; 4Department of Physics, Faculty of Science, Sivas Cumhuriyet University, Sivas 58140, Turkey; ekasap@cumhuriyet.edu.tr; 5EIA-Física Teórica y Aplicada, Universidad EIA, Envigado PC 055428, Colombia; rrestre@gmail.com; 6Grupo de Materia Condensada-UdeA, Facultad de Ciencias Exactas y Naturales, Instituto de Física, Universidad de Antioquia UdeA, Calle 70 No. 52-21, Medellín AA 1226, Colombia; jalexander.gil@udea.edu.co (J.A.G.-C.); alvaro.morales@udea.edu.co (A.L.M.)

**Keywords:** truncated conical quantum dots, exciton states, donor-impurity states, applied magnetic field, type II quantum dots

## Abstract

Using the effective mass approximation in a parabolic two-band model, we studied the effects of the geometrical parameters, on the electron and hole states, in two truncated conical quantum dots: (i) GaAs-(Ga,Al)As in the presence of a shallow donor impurity and under an applied magnetic field and (ii) CdSe–CdTe core–shell type-II quantum dot. For the first system, the impurity position and the applied magnetic field direction were chosen to preserve the system’s azimuthal symmetry. The finite element method obtains the solution of the Schrödinger equations for electron or hole with or without impurity with an adaptive discretization of a triangular mesh. The interaction of the electron and hole states is calculated in a first-order perturbative approximation. This study shows that the magnetic field and donor impurities are relevant factors in the optoelectronic properties of conical quantum dots. Additionally, for the CdSe–CdTe quantum dot, where, again, the axial symmetry is preserved, a switch between direct and indirect exciton is possible to be controlled through geometry.

## 1. Introduction

For decades, low dimensional systems have been some of the most widely investigated objects in semiconductor physics because of their interesting properties and applications, particularly their quantum confinement effects, which have led to different ‘recipes’ for designing novel semiconductor materials for optoelectronic devices [1,2,3,4]. Researchers have been attracted to the theoretical analysis of the effect of the quantum confinement on the impurity energies in various nanostructures, such as quantum wells [5,6,7], quantum-well wires [8], and quantum dots (QDs) [9,10,11]. In particular, in QDs, which are formed when there is a difference in the energy gap between the materials to be used [12,13], the charge carriers (electrons and holes) are subjected to three-dimensional confinement, resulting in a discrete energy spectrum for the charge carriers, and the systems are sensitive to nanoscale changes in geometry and composition, which generate important modifications in semiconductor properties, such as optical, mechanical, electrical, and thermal. This character is similar to that observed in atoms, but with the advantage that, for a QD, the spectrum is adjustable with changes in geometry or by applying external effects, such as electric fields, magnetic fields, and nonresonant intense laser fields. Researchers have also looked into the presence of shallow donor and acceptor impurities and neutral and charged excitons, which generate changes in the confinement of charge carriers due to attractive or repulsive effects.

In the study of QDs, one relevant characteristic involves the geometrical shape of the nanostructure. Researches on semiconductor QDs have included different morphologies: pyramidal, spherical, and lens-shaped nanostructures. In the last three decades, different geometries have been intensively studied [12,13,14,15,16,17]. The sizes and shapes of these quantum systems are shown to have more predominance in the properties of semiconductors than their composition [18]. Among these structures, there is special interest in cylindrical QDs [19,20,21]. These, depending on their radius and height ratios, can be represented as (i) 1D-systems, called quantum-well wires, in which the height of the cylinder is much larger than the radius of the structure; and (ii) 2D-systems, called quantum-wells. More recently, the growth and study of QDs with nanocone type structures [22], which have fascinating physical properties for the development of new technologies, have also been possible. As a result of these investigations, nowadays, it is well known that morphologies of nanostructures can manifest unique physical properties of the material [23], for example, cone-shaped QDs can be quantum wires, quantum wells, or QDs, as a function of the structure height and solid angle at the top [24,25].

Nanowires and nanocones have natural abilities to capture light, so their applications in creating optoelectronic devices, such as solar cells and photodetectors, are promising. Currently, quantum wires (nanowires) and cone-shaped QDs (nanocones) are recognized as promising candidates for the next generation of nanoscale devices [26,27,28,29]. A strong dependence of light absorption on geometry has been demonstrated, for some time, via comparisons between conical and cylindrical nanowire properties [30]. The optical and electronic properties of tapered QDs, including the effects of electric and magnetic fields and donor impurities, have also been studied in different works [31,32].

Analytical solutions of the Schrödinger equation are possible in very limited situations [33], for that, several numerical methods are found in the literature, solving effective mass differential equations and modeling different properties of semiconductor QDs. However, these analytical solutions are used in various works, to show the validity of numerical calculations. The different variants of diagonalization, variational calculus, finite differences, and finite elements are among the methods used. The computation time of the numerical calculation can be long if the codes are not optimized or if there is a need to obtain results for some parameter that is very susceptible to small changes, such as for the magnetic field. The finite element method (FEM) has been used to model QDs since the early 1990s [34], and at present, several studies dealing with electronic structure can be mentioned: optical, structural, impurity, transport, and deformation effects in QDs [35,36,37]. In [38], research on binding energy and susceptibility for cylindrical and spherical QDs, under a different kind of confinement potential, was carried out. In recent works, exciting applications surrounding QDs have also been conducted. Due to their outstanding optical properties, QDs can be used for cancer cell imaging [39,40]. A review article on biomedical and drugs administration mediated by QDs was developed in [41]. CdSe and CdTe QDs are candidates for several applications, such as memory and spintronic devices [42].

Type-II QDs come from a combination of two semiconductors, where the alignment of the energy gaps gives rise to the confinement of electrons and holes in different regions of space. In this case, while one of the semiconductors behaves as the region of the well for the electron and the barrier for the hole, the situation is reversed in the other semiconductor. Thus, the second semiconductor behaves as the barrier region for the electron and the well region for the hole. In regard to core–shell QDs, formed, for example, by the combination of CdTe and CdSe, it is observed that: (i) the CdTe material is the barrier/well region for the electron/hole and (ii) the CdSe material is the well/barrier region for the electron/hole. In Reference [43], and the references included therein, the multiple applications of colloidal QDs are presented in ample detail. The following, for example, are discussed: biophotonics applications, applications in nanomedicine, pharmacokinetics and biodistribution, in vitro and in vivo toxicity, quantum confinement effects, core/shell architectures, tunability in the biological transparency window, opportunity to introduce tunable plasmonic features, doping to achieve enhanced emission from dopant states, and magnetic doping to introduce magnetic imaging capability.

In this article, we are interested in studying the electron, shallow donor impurity, and heavy hole exciton states for two kinds of conical QDs: (i) shallow donor impurity states in truncated conical shaped GaAs-(Ga,Al)As QDs, which can be modeled through a Coulomb interaction, in the simple model of a hydrogenic atom, and considering the effects of an externally applied magnetic field. The magnetic field and impurity center are considered to preserve the axial symmetry of the system; (ii) CdSe–CdTe core/shell QDs without magnetic field and impurity effects. Once the wavefunctions and energies for the electron and hole are available, in the presence or absence of impurity, the interaction between both carriers is calculated using the Coulomb integral, together with a first-order perturbative model. We carry out calculations for different donor impurity positions along the symmetry-axis, considering the effects of the magnetic field and the side of the structure (i). Finally, the overlap integral is reported, information that is key to understanding the behavior of the binding energies for each configuration in (i) and (ii). The solution of the differential equations is obtained by applying the FEM. The paper is organized as follows: Section 2 contains the theoretical framework; Section 3 is devoted to the results and corresponding discussion; finally, in Section 4, we present the main conclusions.

## 2. Theoretical Model

An illustrative scheme of the GaAs-AlGaAs QD under study is shown in Figure 1. Figure 1a is an axis-symmetric representation of the problem (φ=0, with φ the azimuth angle), where the dimensions of the QD bases radii ($R_{1}$ and $R_{2}$) and the QD height (*h*) are shown. Two effects on the structure have been taken into account: (i) a static magnetic field $\vec{B}$, applied in the *z*-direction, and (ii) the presence of a shallow donor impurity at different positions along the same *z*-direction—$\left(0,z_{i}\right)$. In Figure 1b a three-dimensional view of the system obtained from the rotation of Figure 1a around the *z*-axis is shown. In Figure 1c,d are shown the schematic views of the CdTe–CdSe and CdSe–CdTe truncated core–shell QDs without magnetic fields and impurity effects. We have two specific cases: in Figure 1c, a system of CdTe (core) and CdSe (shell), and in Figure 1d, the materials are reversed, i.e., CdSe (core) and CdTe (shell). The ξ-parameter in Figure 1c is the thickness of CdSe, whereas in Figure 1d it is the thickness of CdTe.

Using the effective mass and parabolic bands approximations, with Dirichlet boundary conditions at the outer edges of the barrier matrix, and the Ben Daniel–Duke conditions at the QD and surrounding barrier-matrix interface (see Figure 1b), the Schrödinger equation for an electron (or heavy hole) confined in the structure under the effect of an applied magnetic field, in the *z*-direction, and in the presence of a shallow donor impurity, can be written, in Cartesian coordinates, in the form:(1)$$\left[\frac{1}{2\;m_{j}^{*,c}}{\left(\hat{p}-q\;\vec{A}\right)}^{2}+V_{j}\left(x,y,z\right)+\frac{\kappa \;q^{2}}{4\;\pi \;\varepsilon_{0}\;\varepsilon_{r}\;r}\right]\;\psi \left(x,y,z\right)=E\;\psi \left(x,y,z\right)\;,$$
where $r=\sqrt{x^{2}+y^{2}+{\left(z-z_{i}\right)}^{2}}$ is the electron impurity (hole impurity) distance, $\hat{p}=-i\;\hbar \;\vec{\nabla }$, $m_{j}^{*,c}$ is the electron or heavy hole effective mass (j=e,h for the electron and heavy hole, respectively, and c=w/b indicates the dot/barrier material), q=−e,+e are the electron and hole charges, respectively, *e* being the elemental charge, κ=−1 for the electron, whereas κ=+1 for heavy hole, and $\vec{A}=-\frac{B}{2}\;\left(y\;\hat{i}-x\;\hat{j}\right)$ is the vector potential associated to the applied magnetic field, where $\vec{B}=\vec{\nabla }\times \vec{A}$ comes from the symmetric gauge. $V_{j}$ is the structural potential, which is zero in the dot region and $V_{j}^{0}$ in the barrier material. Additionally, $\varepsilon_{0}=8.85\times 10^{-12}$ C${}^{2}$/(N m${}^{2}$) and $\varepsilon_{r}$ is the static dielectric constant. The image charge effects have been ignored.

Expanding the first term in Equation (Equation 1) and using the azimuthal symmetry condition of the structure, it is possible to consider in cylindrical coordinates a solution of the type $\psi \left(x,y,z\right)=\psi \left(\rho ,\varphi ,z\right)=R\left(\rho ,z\right)\;e^{i\;l\;\varphi }$. Consequently, the R(ρ,z) function satisfies the differential equation
(2)$$\left[-\frac{\hbar^{2}}{2\;m_{j}^{,c}}\nabla^{2}+V_{j}^{c}\left(\rho ,z\right)+\frac{\hbar^{2}\;l^{2}}{2\;m_{j}^{,c}\;\rho^{2}}-\frac{q\;\hbar \;B\;l}{2\;m_{j}^{,c}}+\frac{q^{2}\;B^{2}\;\rho^{2}}{8\;m_{j}^{,c}}+\frac{\kappa \;q^{2}}{4\;\pi \;\varepsilon_{0}\;\varepsilon_{r}\;r}\right]R\left(\rho ,z\right)=E\;R\left(\rho ,z\right)\;,$$
where l∈Z is the azimuthal quantum number and $\nabla^{2}$ is the ρ- and *z*-dependent two-dimensional Laplacian operator.

In previous studies reported in the literature, we analyzed the hydrostatic pressure and size effects on the conduction electron *g*-factor in GaAs-Ga${}_{1-x}$Al${}_{x}$As quantum wells under magnetic field. Clearly, the results have shown that, given the quantum confinement, the *g*-factor changes drastically with the size of the structure concerning its value in bulk and that hydrostatic pressure is an excellent tool by which the *g*-factor magnitude, and even the sign, can be manipulated [44,45]. In this article, we focus our interests on the effects of impurities and their positions, the shape and size of the QDs, as well as the type of coupling between the materials that make up the heterostructure (type-I and type-II heterostructures) for electrons, holes, and excitons confined in conical QDs. We have omitted the Zeeman effect despite considering magnetic field effects, which gives rise to a fine splitting of the energy level structure.

Once the one particle electron and heavy hole ground state wavefunctions ($\psi_{e}^{1}\left({\vec{r}}_{e}\right)$ and $\psi_{h}^{1}\left({\vec{r}}_{h}\right)$, respectively) are obtained, we can proceed to compute the excitonic contribution from the interaction between the two charges. In a first order perturbative approximation, the Coulomb integral magnitude reads
(3)$$C_{eh}=\frac{q^{2}}{4\;\pi \;\epsilon_{0}\;\epsilon_{r}}\int_{\Omega_{h}}\int_{\Omega_{e}}\frac{{\left|\psi_{e}^{1}\left({\vec{r}}_{e}\right)\right|}^{2}{\left|\psi_{h}^{1}\left({\vec{r}}_{h}\right)\right|}^{2}}{\left|{\vec{r}}_{e}-{\vec{r}}_{h}\right|}dV_{e}\;dV_{h}\;,$$
where $dV_{e}=\rho_{e}\;d\rho_{e}\;dz_{e}\;d\varphi_{e}$ and $dV_{h}=\rho_{h}\;d\rho_{h}\;dz_{h}\;d\varphi_{h}$ are the volume differentials in cylindrical coordinates for the electron and hole, respectively. In Equation (Equation 3), $\Omega_{h}$ and $\Omega_{e}$ indicate the volume of the cylinder represented in Figure 1b, for the hole and electron, respectively, whose radius and height are $L_{x}$ and $L_{y}$, respectively. Since we are only interested in the magnitude of the Coulomb interaction, in Equation (Equation 3), we have omitted the negative sign of the electrostatic energy.

Because of the azimuthal symmetry, it is possible to write the angular part of Equation (Equation 3) analytically. This reduces the integral from 6 to 4 variables:(4)$$C_{eh}=\frac{q^{2}}{4\;\pi \;\varepsilon_{0}\;\varepsilon_{r}}\int_{S_{h}}\int_{S_{e}}{\left|\psi_{e}^{1}\left(\rho_{e}\;,z_{e}\right)\right|}^{2}{\left|\psi_{h}^{1}\left(\rho_{h}\;,z_{h}\right)\right|}^{2}\;\left[\frac{8\pi \;K\left(\frac{r_{p}}{1+r_{p}}\right)}{r\;\sqrt{1+r_{p}}}\right]\;dV_{e}^{{}^{\prime }}\;dV_{h}^{{}^{\prime }}\;,$$
where $r=\sqrt{{\left(\rho_{e}-\rho_{h}\right)}^{2}+{\left(z_{e}-z_{h}\right)}^{2}}$, $r_{p}=\frac{4\rho_{e}\;\rho_{h}}{r}$, K(x) is the complete elliptic integral of the first kind, $dV_{e}^{{}^{\prime }}=2\;\pi \;\rho_{e}\;d\rho_{e}\;dz_{e}$, and $dV_{h}^{{}^{\prime }}=2\;\pi \;\rho_{h}\;d\rho_{h}\;dz_{h}$. In Equation (Equation 4), the expression inside the squared parenthesis comes from the double angular integral of the inverse of electron–hole distance and $S_{h}$ and $S_{e}$ correspond to the large rectangle area in Figure 1a. Note that in Equation (Equation 3) and due to the azimuthal symmetry of the system, the electron and heavy hole ground state wavefunctions are independent of the $\varphi_{e}$ and $\varphi_{h}$ coordinates, respectively.

We should note that, in this article, we followed the same strategy previously used in conical QDs subjected to combined effects of electric and magnetic fields [46]. In Reference [46], the calculations of the electron and hole states using the FEM were contrasted with results obtained through a diagonalization process, with an orthonormal basis composed of a product between Bessel and sinusoidal functions, with a coincidence of up to 0.1 meV. Additionally, comparisons were made with a previous more elaborated model [47], which considers the 8-bands $\vec{k}\cdot \vec{p}$ theory and configuration interaction with deviations only by a few percent. In Reference [46], and in this work, the electron–hole Coulomb interaction obtained via a perturbative calculation was corroborated with variational calculations, considering two types of trial functions: (i) a hydrogenic-like function with one variational parameter [48,49]; and (ii) a trial function constructed by the product of two independent Gaussian functions. In this second case, two variational parameters were used to describe the radial problem and the problem along the axial direction [50,51] separately. The results obtained by the perturbative calculation coincided with the variational ones in the margin of 97% for the binding energy.

A quantity that allows complementing the exciton analysis is the overlap integral between the electron and heavy hole ground states, whose calculation is obtained in cylindrical coordinates using the expression
(5)$$I_{eh}=4\;\pi^{2}\;{\left|\int_{S}\psi_{e}^{1}\left(\rho ,z\right)\psi_{h}^{1}\left(\rho ,z\right)\rho \;d\rho \;dz\;\right|}^{2}\;,$$
where in the previous equation, the electron and heavy hole are located simultaneously at the same place, (ρ,z) of the large rectangle in Figure 1a with area *S*.

The wavefunctions and corresponding energies associated with Equation (Equation 2) have been obtained by implementing the FEM [52,53,54,55,56]. Within the COMSOL-Multiphysics licensed software [54,55,56], a user-controlled mesh was chosen in order to achieve greater control over discretization. Since the quantum states of interest correspond to the location of the charge carrier in the GaAs QD, three refinements have been generated in that region of the system. As a result, the number of evaluation nodes in the entire mesh is 7005. Thanks to the mesh adaptation, there are 11,328 triangles in the GaAs QD region and 2513 in the AlGaAs matrix. On the border between the QD and the matrix, there are 217 nodes. Other general characteristics of the mesh are the maximum and minimum element size of 1.85 nm and 0.00625 nm, respectively. In an 8th generation Intel core i7 processor, the computation time for calculating energies in fixed parameters is approximately 8 s. For the numerical calculation of Equation (Equation 4), a Fortran 77 code was used. From the results obtained in COMSOL-Multiphysics, the values of the ground state wavefunction were exported in a regular mesh, in terms of the (ρ,z) coordinates, taking as input of these coordinates the same values for electron and hole. Since we have values of the wavefunctions at discrete points of the coordinates, the integral in Equation (Equation 4) is converted to a Riemann sum. Due to the azimuthal symmetry of the system, the integration corresponding to the φ-angular coordinates was obtained by using elliptic integrals.

## 3. Results and Discussion

### 3.1. Electron and Hole Spectra in GaAs-Al${}_{0.3}$Ga${}_{0.7}$As Truncated Conical Quantum Dot under Donor Impurity and Static Magnetic Field Effects

In Section 3.1 and Section 3.2 the parameters we will use are: $m_{e}^{*,w}=0.067\;m_{0}$ (where $m_{0}$ is the free electron mass), $m_{e}^{*,b}=0.092\;m_{0}$, $m_{h}^{*,w}=0.51\;m_{0}$, $m_{h}^{*,b}=0.57\;m_{0}$, $V_{e}^{0}=0.262$ eV, $V_{h}^{0}=0.174$ eV, and ε=13 [46,57].

In this subsection, the impurity and static magnetic field effects on GaAs-Al${}_{0.3}$Ga${}_{0.7}$As truncated conical QD are explored.

In Figure 2, the electron and heavy hole energy spectra in a GaAs-Al${}_{0.3}$Ga${}_{0.7}$As truncated conical QD are shown as a function of the lower base radius ($R_{2}$), leaving fixed the upper base radius ($R_{1}$) and the dot height (*h*). Three different positions for the donor impurity were considered according to the color code. In order to interpret the results, the ground state level for impurity absence was also plotted. Figure 2a,b are for the ground states of the electron and hole, respectively. Figure 2c is the corresponding binding energy for impurity associated with the electron. This fact allows having a change in geometry from a conical QD ($R_{2}=0$), through a cylindrical QD ($R_{2}=R_{1}=10$ nm), and reaching a truncated conical QD, as shown in Figure 1b. The results are given with consideration of donor impurity and without applied magnetic field effects. In general, it is observed that by increasing $R_{2}$, there is a decrease in the confinement effect for both charge carriers due to the increase in the volume of the structure. This is reflected in a systematic decrease in all energy levels with $R_{2}$. The ground state exhibits a higher rate of decrease in energy values in the range $0<R_{2}<10$ nm. This behavior is explained by the fact that for these small $R_{2}$ values, the vertex region expels the carriers’ wavefunction towards the base of radius $R_{1}$. When $R_{2}$ increases, the ground state rapidly shifts its maximum probability towards the lower dot region, a condition that occurs until the formation of the cylindrical QD ($R_{2}=10$ nm). When the value of $R_{2}$ continues to increase, the ground state, which tends to show its maximum probability density at the center of the QD (away from the edges), does not show drastic changes, which results in energy with a low rate of decline.

The most considerable differences between the electron and the hole spectra occur in the magnitudes of their energies, a situation that is typical of the potential barriers associated with each particle and of their corresponding effective masses. A donor impurity is located in three different positions along the *z*-axis: $z_{i}=0$, $z_{i}=7.5$ nm, and $z_{i}=15$ nm. Compared with the energy without the impurity, the difference related to the interaction between the two types of carriers and the impurity is remarkable. On the one hand, the electron experiences attraction towards the impurity reflected in a spectrum shift towards lower energies. In contrast, the heavy hole is modeled with a repulsive interaction with the donor impurity (see the Equation (Equation 1)), which causes an energy increase for each impurity position. In Figure 2a, the ground state with an impurity at $z_{i}=0$, for $R_{2}$ values close to zero, is the one with the smallest energy redshift in comparing with the other two positions of the impurity, due to the existence of a competition between the geometrical confinement, which causes the wavefunction to be concentrated in the upper part of the QD, and the attractive effect of the impurity, which is responsible for the impurity shifting towards the lower structure region. Clearly, for the ground state, the Coulomb interaction effect is less significant concerning the geometric effect. It should be noted that the presence of the impurity does not influence the degeneracy associated with azimuthal symmetry.

The ground state binding energy ($E_{b}$) for the same three impurity positions is obtained by the difference $E_{b}=E_{1}-E_{1}^{i}$, where $E_{1}$ is the electron ground state energy in the absence of impurity center (κ=0) and $E_{1}^{i}$ is the corresponding one but in the presence of the impurity (κ=−1). For $z_{i}=0$, the binding energy is increasing in the range $0<R_{2}<7$ nm. In this $R_{2}$-regime, the maximum probability density occurs in the upper part of the QD (that is, near the surface of radius $R_{1}$) because the electron tends to be in the region of least confinement; that is, in the region of greater local volume and away from the edges of the cone. With the appearance of the impurity at $z_{i}=0$, there is a systematic decrease in the electron impurity distance as $R_{2}$ increases due to the decrease in the repulsive effect associated with the potential barriers present in the apical point. This results in an increase in the Coulomb interaction and consequently in the binding energy. Once the cylindrical shape of the QD is obtained ($R_{2}=10$ nm) and $R_{2}$ continues to increase from there, it is observed an effective reduction of the confinement effect due to the systematic increase in the volume of the structure. In this case, the binding energy is mainly associated with the Coulomb interaction, and the geometric confinement associated with the structure is transformed into a perturbative effect that decreases with the increase of $R_{2}$. For the case in which the impurity is located in the half-height of the QD, $z_{i}=7.5$ nm, the binding energy is an increasing function in the range $0<R_{2}<2.3$ nm, which is associated with the fast saturation and weak effect of the potential barriers located at the apical point of the inverted truncated cone ($R_{2}<R_{1}$). When $z_{i}=15$ nm, the $E_{b}$ is a monotonically decreasing function of $R_{2}$. In this case, the increase of $R_{2}$ implies a constant displacement of the probability density maximum from the top of the structure towards the half-height region, or even below it, with a permanent increase of the mean electron impurity distance. This is reflected in a drop in electrostatic interaction and consequently in binding energy.

In Figure 3, the energy results are presented as a function of the applied magnetic field for l=0 and l=1, both for the electron, Figure 3a, and the heavy hole, Figure 3b, confined into a truncated conical-shaped GaAs-Al${}_{0.3}$Ga${}_{0.7}$As QD. For κ=0, the electron ground state maintains its symmetry over the entire range of the calculated magnetic field, which corresponds to an *s*-like state with l=0. In the case of the heavy hole, there are multiple crossovers for excited states since one set loses degeneration accompanied by an increase in energy, and the other set of states, which was part of the degeneration in the absence of a magnetic field, goes towards lower energies. This behavior can be observed, for example, for the first excited state in Figure 3b with an impurity at $z_{i}=7.5$ nm; the first excited state (l=1) goes towards lower energies, and on the contrary, the ground state (l=0) goes towards higher energies.

Considering the presence of a donor impurity, which is located at three different positions along the *z*-axis: $z_{i}=0$, $z_{i}=7.5$ nm, and $z_{i}=15$ nm, the combined effect of the applied magnetic field and the impurity allows for shifting the energy levels and generating a rise in the degeneracy for the states with l≠0. The one particle electron’s ground state has its maximum probability density along the *z*-axis, which coincides with the impurity position. In this case, the magnetic field generates additional confinement towards the *z*-axis; thus, effectively reducing the electron impurity distance, thereby enhancing the Coulomb interaction. The greatest effects on the energies of the system with interaction occur for the impurity located at the QD’s half-height, Figure 3c-green color, since the influence of the Coulomb interaction occurs in all directions of space. For the heavy hole, the repulsive effect of the donor impurity generates a quantum ring-like behavior, which becomes evident with the ground state’s oscillations when the magnetic field is turned on. For example, in Figure 3, the ground state corresponds to l=0 for B=0 while for B=15 T the ground state occurs with l=+1. It is evident from that, for impurities in the lower and upper bases of the structure, the heavy hole ground state continues to present the oscillatory character with the magnetic field, which means changes in the quantum numbers corresponding to the ground state when the magnetic field increases. However, the repulsion generated by the potential barriers reduces the manifestation of this effect. In Figure 3c, the ground state binding energies for the confined electron are presented as a function of the applied magnetic field for three considered donor impurity positions, $z_{i}=0$, $z_{i}=7.5$ nm, and $z_{i}=15$ nm. The results were obtained by subtracting from the ground state energy in the black color in Figure 3a, the corresponding ground state energies to the three impurity positions. Although the magnetic field decreases, the expected value of $\rho =\sqrt{x^{2}+y^{2}}$; that is, it compresses the wavefunction towards the *z*-axis, when the impurity is located at the upper or lower face of the QD, the variation of the binding energy increases to a low rate. For impurities located at the QD’s half-height, $z_{i}=7.5$ nm, the most significant interaction with the electron is reached. With the increase in the magnetic field, there is a smaller distance between the electron and the impurity, obtaining higher binding energy. In the latter case, as mentioned above, the Coulomb interaction has a quasi-3D symmetrical character, except for the variations that the structure shape induces at the boundaries between the QD and the surrounding material.

Two sets of panels for the electron and heavy hole ground state wavefunction (WF) are shown in Figure 4. There are cuts in the first, third and fourth rows for the y=0-plane. A first set corresponds to the electron WFs, Figure 4a–d and the second one, Figure 4e–l for hole WFs. The real and imaginary parts of the heavy hole WFs correspond to Figure 4e,f,i,j and Figure 4g,h,k,l, respectively. The second set of cuts is for the plane z=7.5 nm, Figure 4c,d,i,l and the two columns are for two different values of the applied magnetic field. With a ground state of *s*-like symmetry, the electron is located fundamentally in the QD center and independently of the applied magnetic field’s value, but the strength of confinement is stronger with the magnetic field. This distribution of the WF is the result of the attraction towards the impurity center. When the magnetic field is applied, it is evident that the electron has a more significant location towards the axial axis. Note that the *s*-like symmetry of the electron ground state is maintained with the magnetic field’s inclusion, even for the highest value of B=20 T used in this work. The real part of the heavy hole WF (Figure 4e,f,i,j) shows that the WF has the ring’s symmetry in the absence of a magnetic field since the donor impurity generates electrostatic repulsion on the hole. When the magnetic field is included, there are changes in the symmetry of the heavy hole ground state (evolving from l=0 for B=0 up to l=+1 for B=20 T; (see the row 5 in Figure 4)). These results are in agreement with the ground state of Figure 3b. For B=20 T, there is complementarity between the WF’s real and imaginary components (see panels (j) and (k)). This means that the sum of the two components’ squares generates a probability density that shows ring-shaped symmetry, as occurs in the absence of a magnetic field, see Figure 4i.

### 3.2. Exciton States in GaAs-Al${}_{0.3}$Ga${}_{0.7}$As Truncated Conical Quantum Dot under Impurity and Static Magnetic Field

This subsection is dedicated to studying the excitonic contribution in GaAs-Al${}_{0.3}$Ga${}_{0.7}$As truncated conical QD considering the effects of shallow donor impurity and static applied magnetic field.

Figure 5 shows the results for the electron–hole pair in a truncated conical-shaped GaAs-Al${}_{0.3}$Ga${}_{0.7}$As QD as a function of the $R_{2}$-lower structure radius. In Figure 5a is shown the Coulomb energy calculated by Equation (Equation 4). The results of Figure 5b are for the overlap integral (OI) between the electron and heavy hole ground states. Finally, the Figure 5c,d correspond to the electron and heavy hole *z*-expected value ($\langle z_{e}\rangle$ and $\langle z_{h}\rangle$). From Figure 5a, it can be inferred that, regardless of the presence or not of the donor impurity, with the increase of $R_{2}$ there is a loss of the electron–hole interaction. Note the decreasing character with $R_{2}$ of all the curves, which can be explained by an increase in the expected value of the electron–hole distance—$\langle |{\vec{r}}_{e}-{\vec{r}}_{h}|\rangle$. In the absence of impurity, the electron and hole are located essentially at the same vertical position. As $R_{2}$ goes from zero to 20 nm, the maxima of the electron and hole probability densities shift (with decreasing values of $\langle z_{e}\rangle$ and $\langle z_{h}\rangle$) from the gravity center of an inverted cone towards the gravity center of a truncated cone with a major/minor radius at the bottom/top base of the QD. This fact explains the quasi-overlap and decreasing behavior of $\langle z_{e}\rangle$ and $\langle z_{h}\rangle$ in Figure 5b–d. The increase of the OI with κ=0 is mainly associated with the increase of the structure’s volume. In the case κ≠0, the OI follows the behavior exhibited by the difference between $\langle z_{e}\rangle$ and $\langle z_{h}\rangle$. The OI is maximum when $|\langle z_{e}\rangle -\langle z_{h}\rangle |\to 0$ and is minimum when $|\langle z_{e}\rangle -\langle z_{h}\rangle |$ is maximum. The previously mentioned change in the structure’s geometry also explains the decreasing value of $\langle z_{e}\rangle$ and $\langle z_{h}\rangle$ in Figure 5c,d. Including the impurity at different positions modifies the behavior of $\langle z_{e}\rangle$ and $\langle z_{h}\rangle$. For $z_{i}=0$, $|\langle z_{e}\rangle -\langle z_{h}\rangle |$ increases in the range $0<R_{2}<7$ nm. With the increase of $R_{2}$, the electron moves towards the impurity center; this explains why it goes faster towards lower $z_{e}$-values. In contrast, the hole is subjected to two conditions that generate inverse effects; on the one hand, the decrease in confinement associated with the increase $R_{2}$ implies a fall in $\langle z_{h}\rangle$, but the rate of decrease is lower than that of $\langle z_{e}\rangle$, with a quasi-linear behavior, because on the other hand, the impurity at $z_{i}=0$ repels the hole. We want to highlight that, for $R_{2}=0$, the situation with the highest confinement on the hole–electron pair occurs. The conical part of the QD at z=0 generates extreme repulsion on the electron and hole wave functions making the effects of an impurity located at $z_{i}=0$ almost undetectable. This fact explains the first case of almost equal energies for “without impurity” and $z_{i}=0$. When discussing extreme confinement, the electron and hole wave functions are difficult to deform due to impurity effects when it goes from $z_{i}=7.5$ nm to $z_{i}=15$ nm. This situation explains the second case of energies almost equal to close to 20 meV.

In Figure 6, the same kind of results are reported as in Figure 5, but for fixed QD dimensions and as a function of the applied magnetic field. In Figure 6a, without impurity, the electron–hole Coulomb energy increases. This fact is largely explained by the decrease in the carriers’ separation with the increase in the magnetic field, information extracted from Figure 6d–f for $|\langle z_{e}\rangle -\langle z_{h}\rangle |$, where the OI systematically increases. Due to the opposite effect that the impurity has on the electron and the hole, that is, attraction and repulsion, respectively, lower Coulomb energies are generally observed compared to the case of no impurity. In Figure 6b,c the Coulomb energies for $z_{i}=7.5$ nm and $z_{i}=15$ nm, respectively, from Figure 6a are plotted. The idea of these plots is to emphasize the confinement behavior of both the electron and the hole when symmetry changes of the ground state WFs occur due to the effect of the applied magnetic field. In Figure 6b, for B<8.5 T, the electron and hole ground state WFs are obtained for l=0, and in this case, the imaginary part of the WFs for both carriers is zero. The *s*-type symmetry for the electron and ring symmetry for the hole is clearly identified. For B>8.5 T, the ground state of the hole corresponds to l=1, and complementarity between the real and imaginary parts of the WFs appears, but a ring symmetry for the probability density is maintained. In Figure 6c, the results are similar to those depicted in Figure 6b, but in this case, the transition of the ground state hole WF moves to a higher magnetic field value. The small increase observed in the Coulomb energy for $z_{i}=0$ and $z_{i}=15$ nm, with the magnetic field influence, is because, despite having a lower OI with the increasing of the magnetic field, the WFs are located in a smaller volume, which increases the Coulomb integral value. The change in symmetry of the WFs for the ground state of the hole explains the jumps in the curves of the Coulomb integral for $z_{i}=7.5$ nm and $z_{i}=15$ nm shown in Figure 6a and detailed in Figure 6b,c with the corresponding electron and hole WFs for two particular values of the applied magnetic field.

In order to have a better perspective on the effects of the impurity position, it was varied from the lower base to the upper base of the truncated conical-shaped GaAs-Al${}_{0.3}$Ga${}_{0.7}$As QD, always located along the *z*-axis. This position is controlled by the parameter $z_{i}$. The following graphs are shown in Figure 7: (a) the electron and hole ground state energies; (b) the electron–hole Coulomb energy; (c) the overlap integral; and (d) *z* the average position of the electron and hole along the *z*-axis. Calculations are for $R_{1}=10$ nm, $R_{2}=20$ nm, h=15 nm, and B=0. In Figure 7a, the reverse effect on the electron and the hole of a donor impurity is clearly seen. The electron has an attractive character, and for the hole, it is a repulsive one. When the impurity is in the first half of the height, that is $z_{i}\leq 15$ nm, the electron is attracted to a region of lower confinement, and the hole is being expelled to the region of higher confinement. In the second half of the height, i.e., for $z_{i}\geq 15$ nm, the opposite effect on each charge carrier occurs. Hence, there is increasing energy for the electron and decreasing energy for the hole. The Coulomb energy in Figure 7b decreases up to $z_{i}=8.5$ nm because statistically speaking, the electron and hole are being separated by the donor impurity. From $z_{i}=8.5$ nm, the competition between the impurity effect and the volume for the electron causes the electron to stay in a region that is in a better overlap with the hole, which is in absolute agreement with the OI in Figure 7c. Since the impurity is located on the *z*-axis, the *z*-expected value for each of the particles is evidence from how the impurity modifies the positions of the charge carriers. Note that the Coulomb energy for the impurity at $z_{i}=0$ is higher than the energy for $z_{i}=15$ nm, which is justified by the larger *z*-separation between the two particles at $z_{i}=15$ nm (see Figure 7d). Moreover, note that for $z_{i}=4.75$ nm the combined effects of geometry and impurity make the expected *z*-value the same for electron and hole. At that value of $z_{i}$ the higher Coulomb energy does not occur because it must be taken into account that there are geometry effects at different *z*-planes.

Finally, we should emphasize that, in those situations where the independent particle energies for the electron and the hole are of the same order of magnitude as the Coulomb interaction between both charge carriers, it is clear that the results obtained by means of a first order perturbative calculation should be viewed with some reserve, and only as a guiding information to analyze the physics of the problem in question.

### 3.3. Tuning from Direct to Indirect Exciton in Truncated Conical CdSe–CdTe Core–Shell Quantum Dots

In this section, according with the Figure 1c,d a core–shell system of CdTe–CdSe and CaSe–CdTe quantum dots is studied, in which a transition between spatially direct and spatially indirect exciton occurs. The parameters we will use in this subsection are: (i) in CdTe $m_{e}=0.096\;m_{0}$, $m_{h}=0.40\;m_{0}$, and ε=10.2; (ii) in CdSe $m_{e}=0.120\;m_{0}$, $m_{h}=0.45\;m_{0}$, and ε=10.2 [58]. Additionally, $V_{e}^{0}=0.42$ eV and $V_{h}^{0}=0.57$ eV [58].

Figure 8 presents the characterization of the exciton states related to the case of Figure 1c. As stated in the Section 1, the CdTe behaves as the barrier/well region for electrons/holes in this case, and the CdSe corresponds to the well/barrier region for electrons/holes. When ξ=0, the electron and hole are confined in a truncated conical QD with an infinite potential barrier. When ξ≠0 appears, the potential well for electrons is located in the CdSe region. For small values of ξ (ξ<1.5 nm), the electron remains in the CdTe region due to the strong confinement in the CdSe region. For ξ>1.5 nm, the CdSe volume becomes large enough for the electron to penetrate the region where its potential well is located. At that time, both charge carriers are each located in the regions of their own potential wells, i.e., the hole in CdTe and the electron in CdSe, and a spatially indirect excitonic system appears. The sharp drop in electron energy in Figure 8a is associated with the change of confinement region and the subsequent increase in the volume of the shell region. The slight perceptible variations in the hole’s energy appear when the finite potential barrier located on the lateral face of the structure becomes present. However, from a specific value of ξ, the hole’s energy does not show variations due to the constant volume of the core region as ξ increases. The drop in the Coulomb energy in Figure 8c is explained by the fact that the mean electron–hole distance is increasing with the ξ-parameter. Figure 8d allows identifying the transition from direct to indirect exciton, due to initially the particles are in the CdTe and as ξ increases, the electron shifts to the CdSe region, with the hole always localized at the CdTe region, reaching an indirect exciton.

The characterization of the exciton states related to the case of Figure 1d is shown in Figure 9, where the results are concerning the ξ-parameter. From Figure 1d, it can be seen that by increasing the ξ-parameter, the volume of the CdSe region decreases, which corresponds to the potential well for electrons. Consequently, the energy of the electron levels must grow with the ξ-parameter, as shown in Figure 9a. When ξ→0, the hole is confined in the region of its potential barrier, surrounded by an infinite potential, and in this case, the energies of all confined states (*E*) fulfill the condition $E>V_{h}^{0}=0.57$ eV, as shown in Figure 9b in the regime ξ<1.0 nm. When ξ>1.0 nm, the volume of the shell region becomes large enough for the hole to migrate towards the region of its potential well, that is, towards the CdTe region. The volume of the CdTe, shell region, grows progressively with the ξ-parameter, and with it, the confinement on the hole decreases, which is manifested in a fall in all energy levels, a situation shown in Figure 9b. We can then conclude that the system presents an evolution from a spatially direct exciton with the hole and the electron both located in the CdSe region towards a spatially indirect exciton for large values of ξ-parameter with the electron in the core region, whose volume decreases, and the hole in the shell region, the volume of which increases. The interpretation presented here is confirmed by the ever-decreasing behavior of the Coulomb integral in Figure 9c and the probability densities in Figure 9d.

By size and shape control of the truncated conical QD and external voltage, combined with an externally applied magnetic field, one can design a four-level QD with appropriate energy levels, suitable for controlling the optical bistability and multistability by terahertz signal field. The calculated QD structures that we report here can be the base components in photoconductive THz emitters/detectors, providing critical fundamental and practical applications at the forefront of scientific knowledge (sensors, flexible electronics, security systems, biomedicine, and others).

## 4. Conclusions

In this paper, we have studied the electron and hole states in GaAs-(Ga,Al)As conical-shaped QDs in the presence of an axially located shallow donor impurity under the effects of an externally applied magnetic field and CdSe–CdTe core–shell QDs without impurity and magnetic field effects. The impurity position and the magnetic field direction preserve the axial symmetry of the system. Variations of the geometry were considered, in which case the structures evolve from conical QDs to truncated conical QDs. Calculations were done in the effective mass approximation and considering a parabolic two-band model. The Schrödinger equations were solved via a finite element method with a flexible discretization mesh. The electron–hole interaction was studied with a first-order perturbation approximation. Among the most relevant results of this study, we can cite: (i) the presence of the shallow donor impurity is responsible for a red/blue shift of the electron/hole energies; (ii) the binding energy for the electron impurity system in general decreases with the size of the structure; but with exceptions for specific geometries, the binding energy is an increasing function with the radius of the lower base of the system; (iii) the applied magnetic field is responsible for the hole impurity ground state oscillations; (iv) the binding energy for the electron impurity system is always an increasing function of the applied magnetic field; and (v) a control to tune between direct and indirect exciton by changes of thickness of CdSe or CdTe in the core–shell type QD was reached. In the case of the interaction energy between the electron impurity and hole impurity states, it is observed that, in general, they decrease with the size of the structure. Considering the presence of applied magnetic fields, they present an essentially constant behavior in specific ranges of the magnetic field with jumps associated with oscillations of the hole impurity ground state. The localization of the electron and hole states in the presence of impurity and the overlap integral are essential information to adequately interpret the Coulomb interaction between the electron impurity and hole impurity states.

The QD nanostructures studied here, combined with electric and magnetic field effects, can be the basis for the design of a four-level quantum dot with appropriate energy levels, suitable for controlling the optical bistability and multistability by the terahertz signal field. Likewise, these structures find potential applications in developing highly efficient solar cells and can even be the basis for components in photoconductive THz emitters/detectors. We hope that our research will stimulate future theoretical and experimental investigations about electronic, impurity, and excitonic states in conical QDs, with potential applications in nanoelectronics.

## Figures and Tables

**Figure 1 nanomaterials-11-02832-f001:**
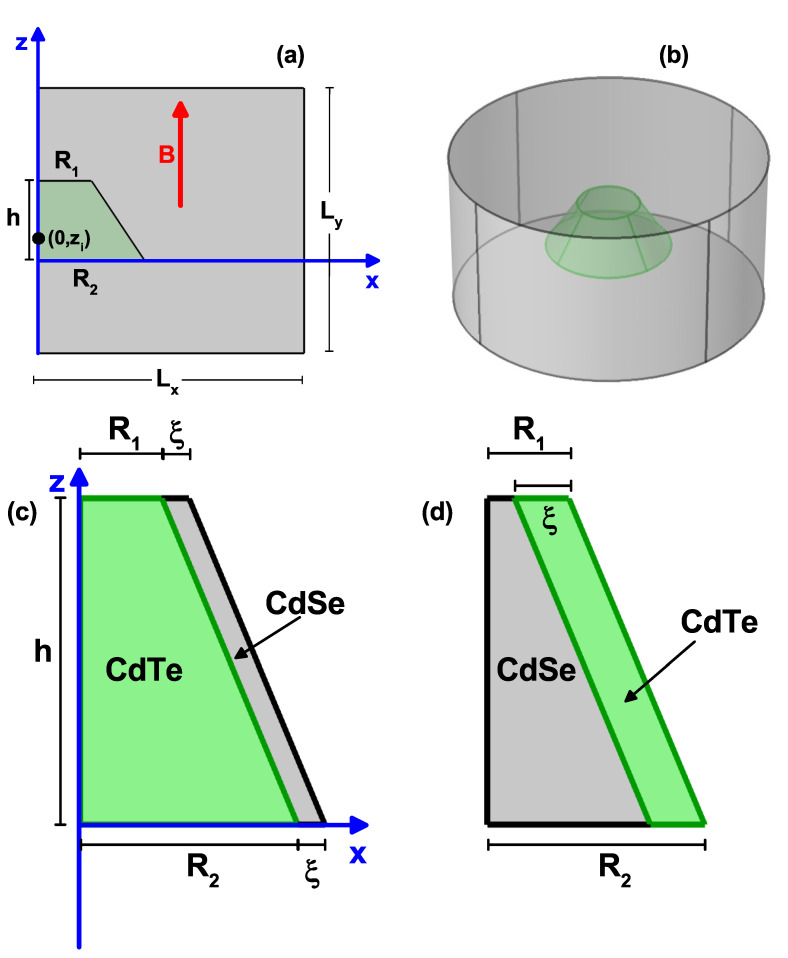
Schematic view of the truncated conical-shaped GaAs-Al${}_{0.3}$Ga${}_{0.7}$As quantum dot, (cone region GaAs, outside the cone AlGaAs). In (**a**) is depicted the φ=0 projection, where the dot dimensions ($R_{1}$, $R_{2}$, and *h*), the shallow donor impurity position $\left(0,z_{i}\right)$, the reference frame, the vertically applied magnetic field, and the dimensions of the large-size square where the open and Dirichlet boundary conditions are applied ($L_{x}=L_{y}=50$ nm), are indicated. The position of the cone half-height coincides with the half-height of the large square, and the reference frame coincides with the base of the cone. In (**b**) is shown the structure obtained by rotating around the *z*-axis the structure represented in (**a**). Schematic view of the truncated conical CdTe–CdSe (**c**) and CdSe–CdTe (**d**) core–shell quantum dots. The ξ-parameter corresponds to the shell width of CdSe (**c**) and CdTe (**d**). $R_{1}$ and $R_{2}$ are the base radii, and *h* is the QD height. In (**c**,**d**), the impurity and magnetic field effects have been neglected.

**Figure 2 nanomaterials-11-02832-f002:**
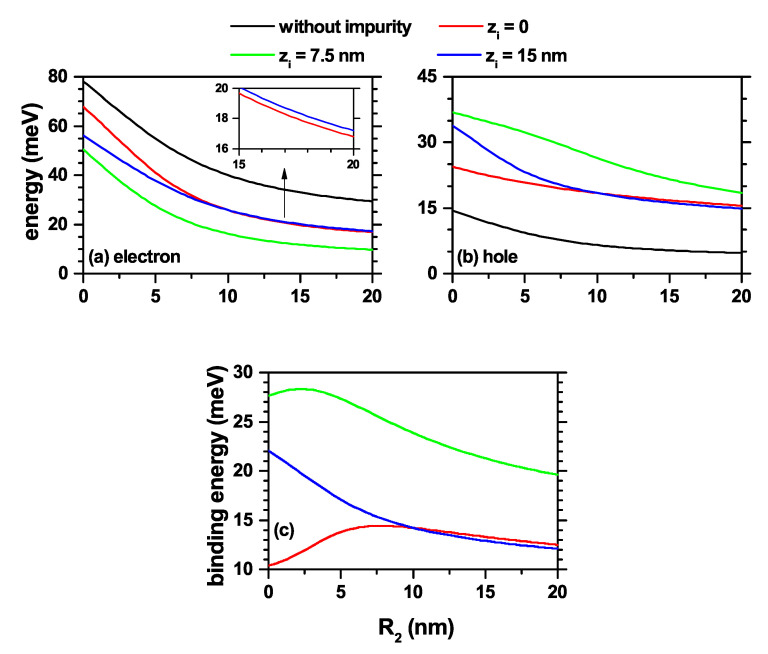
The ground state level for a confined electron/hole (**a**)/(**b**) in a truncated conical-shaped GaAs-Al${}_{0.3}$Ga${}_{0.7}$As quantum dot as a function of the $R_{2}$-lower structure radius and the corresponding biding energy (**c**). Calculations are for $R_{1}=10$ nm, h=15 nm, B=0, with and without impurity effects according to the color code. Three impurity positions were considered: $z_{i}=0$, 7.5 nm, and 15 nm. The inset in Figure 2a shows, with better resolution, that the electron ground state energy for impurities at $z_{i}=0$ and $z_{i}=15$ nm should be different when $R_{2}>R_{1}$, as clearly seen in Figure 2b for the case of the hole.

**Figure 3 nanomaterials-11-02832-f003:**
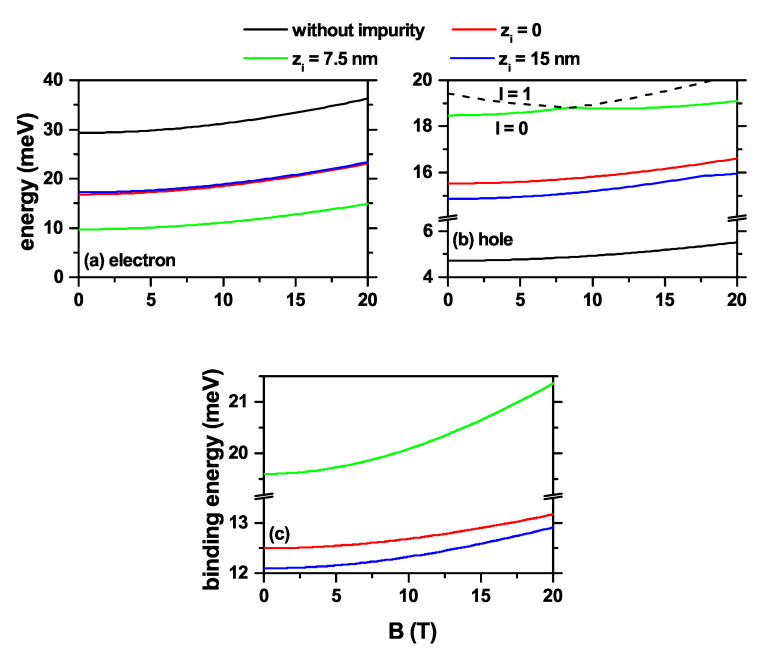
The ground state level for a confined electron/hole (**a**)/(**b**) in a truncated conical-shaped GaAs-Al${}_{0.3}$Ga${}_{0.7}$As quantum dot as a function of the vertically applied magnetic field and the corresponding binding energy (**c**). Calculations are for $R_{1}=10$ nm, $R_{2}=20$ nm, h=15 nm, with and without impurity effects according to the color code. Three impurity positions were considered: $z_{i}=0$, 7.5 nm, and 15 nm.

**Figure 4 nanomaterials-11-02832-f004:**
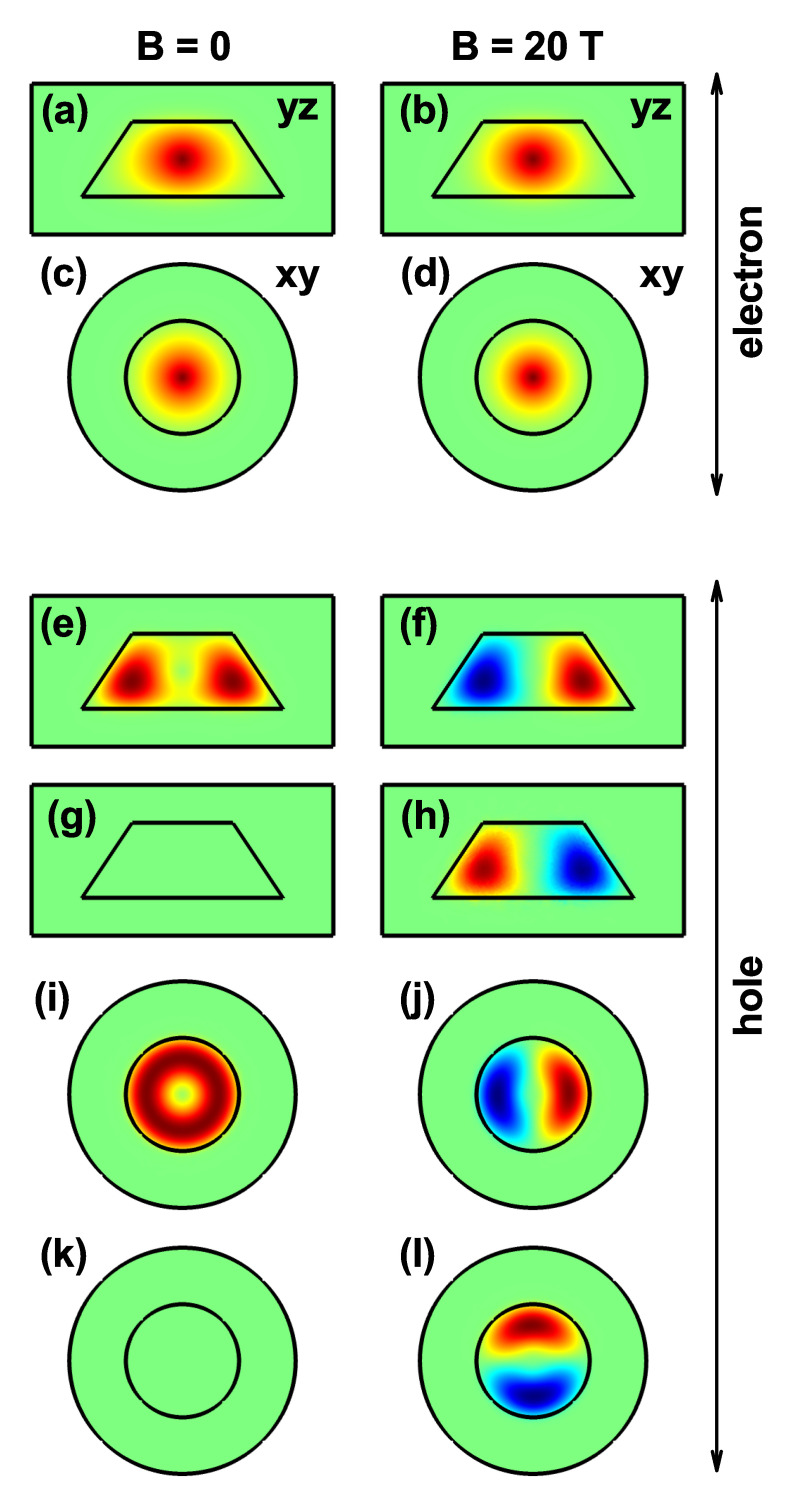
The x=0 (**a**,**b**,**e**–**h**) and z=7.5 nm (**c**,**d**,**i**–**l**) projections of the electron and heavy hole ground state wavefunctions (WF) in a truncated conical-shaped GaAs-Al${}_{0.3}$Ga${}_{0.7}$As quantum dot, for two values of the vertically applied magnetic field (each column corresponds to a fixed value of the magnetic field). The first two rows are for the real part of the electron WF, rows 3 and 5 are for the real part of the hole WF, whereas rows 4 and 6 are for the imaginary part of the hole WF. Calculations are for $R_{1}=10$ nm, $R_{2}=20$ nm, h=15 nm, and with a donor impurity located at $z_{i}=7.5$ nm.

**Figure 5 nanomaterials-11-02832-f005:**
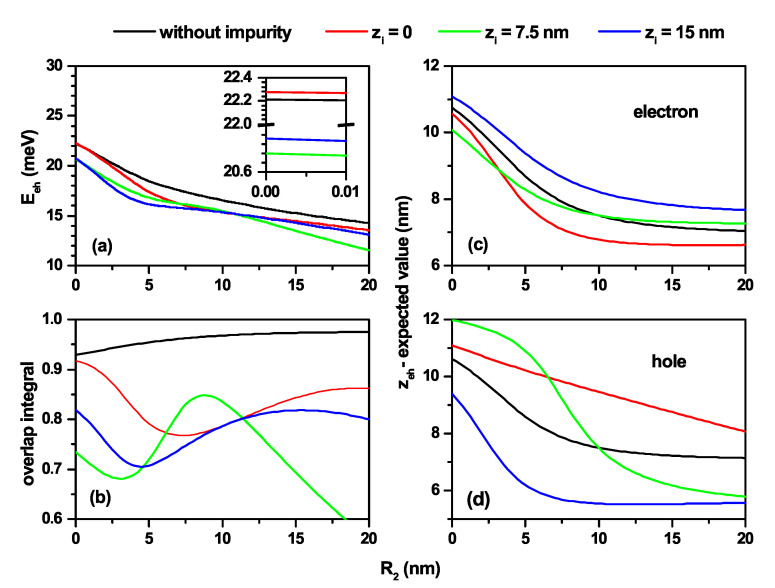
Characterization of the heavy hole exciton states in a truncated conical-shaped GaAs-Al${}_{0.3}$Ga${}_{0.7}$As quantum dot as a function of the $R_{2}$-lower structure radius. (**a**) Electron–hole Coulomb energy, (**b**) overlap integral, and (**c**,**d**) the *z*-average position of the carriers; electron (**c**) and heavy hole (**d**). According to the color code, calculations are without and with a shallow donor impurity, localized at three positions along the *z*-axis. The results are for $R_{1}=10$ nm, h=15 nm, and B=0. The inset in Figure 5a shows in detail the situation $R_{2}∼0$ where it is evident that the four energies are different despite having two almost equal pairs.

**Figure 6 nanomaterials-11-02832-f006:**
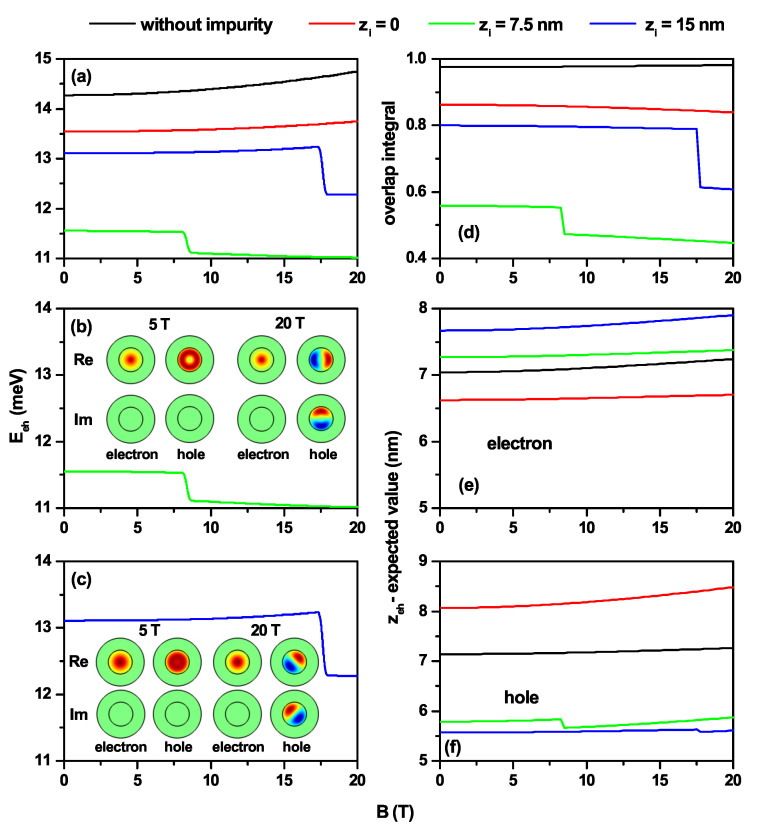
Characterization of the exciton effects in a truncated conical-shaped GaAs-Al${}_{0.3}$Ga${}_{0.7}$As quantum dot as functions of the vertically applied magnetic field. (**a**) e–h Coulomb energy with κ=0 and κ≠0 with three impurity positions, (**b**) e–h Coulomb energy with the impurity at $z_{i}=7.5$ nm, (**c**) e–h Coulomb energy with the impurity at $z_{i}=15$ nm, (**d**) overlap integral, (**e**) $z_{e}$-average electron position, and (**f**) $z_{h}$-average hole position. The results are for $R_{1}=10$ nm, $R_{2}=20$ nm, and h=15 nm. Few electron and hole wavefunctions projections (real and imaginary parts) at the z=7.5 nm plane are shown in panels (**b**,**c**).

**Figure 7 nanomaterials-11-02832-f007:**
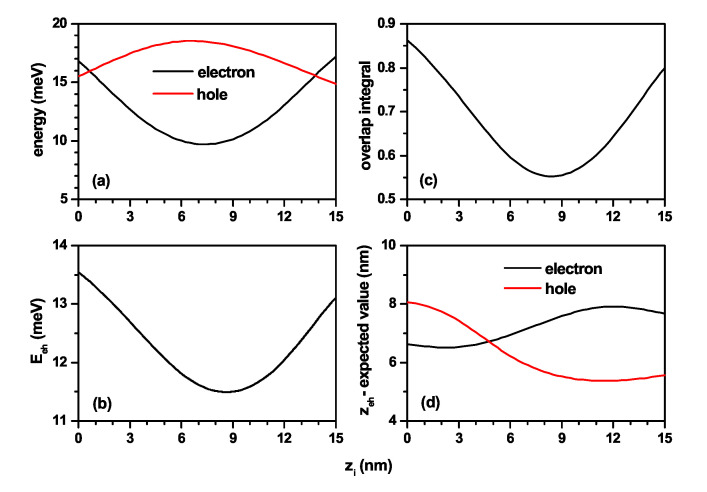
Characterization of the exciton effects in a truncated conical-shaped GaAs-Al${}_{0.3}$Ga${}_{0.7}$As quantum dot as a function of donor impurity position $z_{i}$. (**a**) Ground state for electron and hole; (**b**) electron–hole Coulomb energy; (**c**) overlap integral; and (**d**) *z* expected value for each carrier. Calculations are for $R_{1}=10$ nm, $R_{2}=20$ nm, h=15 nm, and B=0.

**Figure 8 nanomaterials-11-02832-f008:**
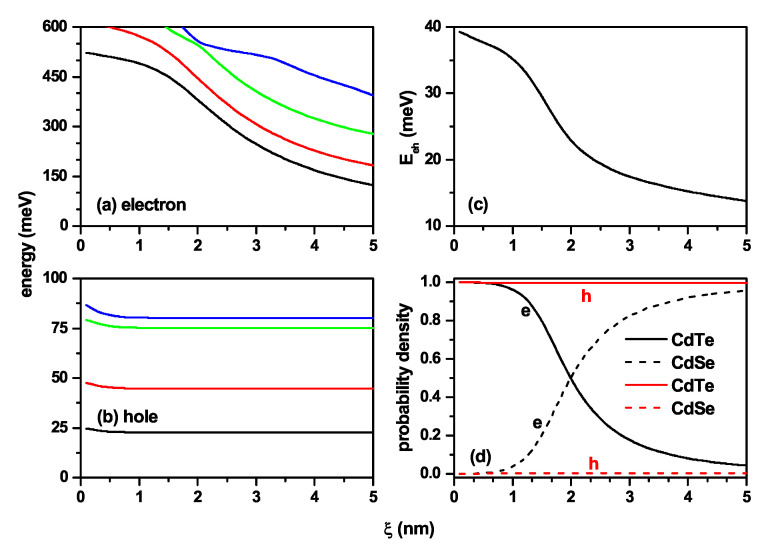
Characterization of the exciton effects in a truncated CdTe–CdSe conical-shaped quantum dot concerning the ξ-parameter. The first fourth energy levels for the electron (**a**) and hole (**b**) are shown. In (**c**), electron–hole Coulomb energy is depicted, whereas in (**d**) are depicted the probability density for both the electron and hole in each material CdTe and CdSe. The calculations are for $R_{1}=3$ nm, $R_{2}=8$ nm, h=12 nm, B=0, and impurity absence.

**Figure 9 nanomaterials-11-02832-f009:**
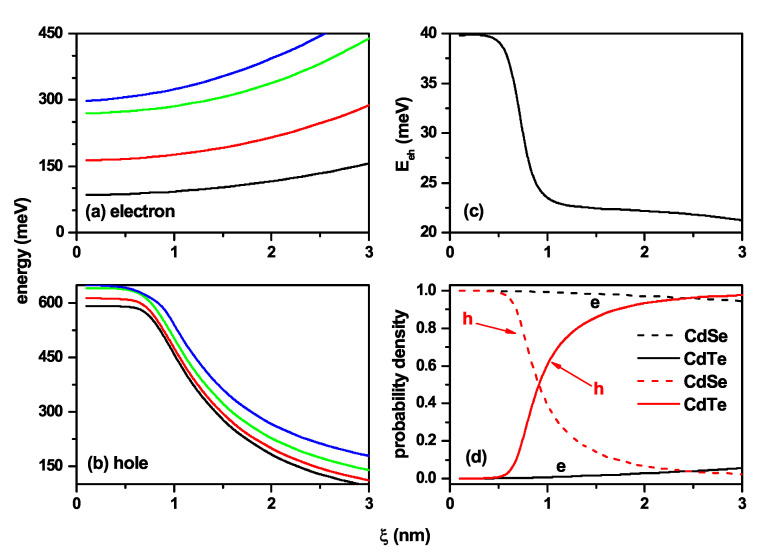
Characterization of the exciton effects in a truncated CdSe–CdTe conical-shaped quantum dot as a function of the ξ-parameter in accordance with Figure 8b. The lowest energy levels for the electron (**a**) and hole (**b**) are shown. In (**c**), the results are for the electron–hole Coulomb energy, whereas in (**d**), the probability density for both electron and hole in each CdSe and CdTe material are presented. The calculations are for $R_{1}=3$ nm, $R_{2}=8$ nm, h=12 nm, B=0, and impurity absence.

## Data Availability

No new data were created or analyzed in this study. Data sharing is not applicable to this article.

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
