# Peer review of "Shallow Donor Impurity States with Excitonic Contribution in GaAs/AlGaAs and CdTe/CdSe Truncated Conical Quantum Dots under Applied Magnetic Field"

_nanomaterials, 2021, doi:10.3390/nano11112832_

Round 1

Reviewer 1 Report

In the work, "Shallow-donor impurity states with excitonic contribution in GaAs/AlGaAs and CdTe/CdSe truncated conical quantum dots under applied magnetic field", the authors consider two kinds of conical quantum dots (QD), and they use the effective mass approximation to study the electron and hole states. The authors investigate the settings by considering an axial impurity at different positions and the effect of the magnetic field, as well.

There are interesting points that could lead to the publication of this work. Nevertheless, instead of being pointed out to motivate the reader, they are hidden in extensive descriptive texts. Also, there is no reference to potential applications and future uses of these findings. Since the publication is submitted to a journal relevant to materials (which is a quite applied subject) such links are necessary.

While the manuscript is in this form, I cannot recommend publication. However,

  • if the length of the manuscript is reduced, giving emphasis to the main findings and their importance
  • if the results are linked to possible applications (experiments, possible material functionalities, etc.),

then publication in “Nanomaterials” could be possible.

A few, indicative comments/suggestions can be found below:

  1. What do the authors mean by “eigenvalues differential equations”? I guess they mean that they use the eigenvalues to solve the differential equation system. However, the way it is written both in the abstract and in the conclusions does not make sense to me. Maybe, the authors could consider not using it as a term.
  2. The manuscript needs to be checked again for inconsistencies and typos. I.e., in line 86 we find: “In Ref. [Review]…”. In lines 137-139 it seems that some notes have been forgotten there.
  3. I understand that this is a theoretical work, and this is fine. However, since the authors study QDs which are highly relevant to applications, they should devote a few lines (maybe in the conclusion and the introduction) to discuss the application relevance and implications of their findings.
  4. In the first model, the authors consider a QD where axial symmetry is preserved, and they consider all the positions of the impurity on this axis. Is this choice only due to mathematical convenience? What are the implications of this symmetry in the behavior of the system? How the results would be modified if the impurity was not placed on the symmetry axis.
  5. In Fig. 2a, it seems that at R2=R1 where the QD becomes cylindrical, the electron energies for z=0nm and z=15nm cross (reasonable and expected). However, it is not clearly explained why, as R2 increases and R2>R1, the two energies still, approximately, coincide. In (b) the situation is clear. Does this hold for larger R2 values? If this a matter of bad resolution, consider adding an inset zooming at the region 15-to-20nm.
  6. In Fig. 5 (a) the “no impurity” and “z=0” seem to begin from the same value. The same holds for the “z=7.5” and “z=15” cases. I’m not sure that the mechanism for this is adequately explained in the text.
  7. As a final comment, which has to do with the presentation of the manuscript, I would recommend using the same axis font (and style) in all figures. Currently, Figs. 2,3,4 have a quite different format from Figs. 5,6,7,8,9.

Author Response

Referee 1:

The Referee:

In the work, "Shallow-donor impurity states with excitonic contribution in GaAs/AlGaAs and CdTe/CdSe truncated conical quantum dots under applied magnetic field", the authors consider two kinds of conical quantum dots (QD), and they use the effective mass approximation to study the electron and hole states. The authors investigate the settings by considering an axial impurity at different positions and the effect of the magnetic field, as well.

There are interesting points that could lead to the publication of this work. Nevertheless, instead of being pointed out to motivate the reader, they are hidden in extensive descriptive texts. Also, there is no reference to potential applications and future uses of these findings. Since the publication is submitted to a journal relevant to materials (which is a quite applied subject) such links are necessary.

While the manuscript is in this form, I cannot recommend publication. However,

Our reply:

We are deeply grateful to the referee as we consider that he/she has made a very careful reading of our manuscript and has presented us with very valuable suggestions. By considering the Referee's suggestions, we believe that our manuscript has been improved in a very important way. We hope that the new version of our article will be seen by the Referee as suitable for publication in the journal Nanomaterials.

The Referee:

If the length of the manuscript is reduced, giving emphasis to the main findings and their importance

If the results are linked to possible applications (experiments, possible material functionalities, etc.),

then publication in “Nanomaterials” could be possible.

Our reply:

We ask the Referee to allow us to explain our reason for the extension of the manuscript and the physical discussions contained therein. In this article, we have wanted to go into a more detailed explanation of the physics in each of the results obtained due to the fact that the journal Nanomaterials, and in this case the special issue to which the article is submitted, has the scope of reaching a population of readers in very diverse areas of knowledge, not only in Physics of semiconductor heterostructures and, with many degrees of training. We have explained the fundamentals of energies, we have used concepts based on wave functions, we have analyzed in detail electrostatic interactions, the mean distance between charge carriers and Coulomb centers, etc. We have tried to address type-I and type-II heterostructures to make the article more self-contained, we have considered the variations in the position of the impurity so that information is obtained that is relevant when the systems are subjected to unintentional doping, we have studied spatially direct and indirect excitonic systems, the lifetime for excitonic states and the overlap integral in the two types of structures. We have tried to make it a complete article where we can record the experience we have obtained during many years of work in this type of physical system. In truth, we have made a very important effort so that the article can be available to experts, but also available to people who are venturing into this type of study since we have thought that the way we do the discussions can help the people starting in this field of research understand the physics of the problems in question.

For all of the above, we ask the Referee to give us the opportunity for the article not to be reduced in length and not to modify the discussions. Of course, we will review the entire content to detect writing, grammar, and language errors. We will take into account all the observations that the Referee makes to us below.

About the second observation that the Referee makes us in this first part, we want to say that at the end of the results and discussion section we have added the following paragraph concerning the possible applications of our quantum dot system:

By size and shape control of the truncated conical QD and external voltage, combined with an externally applied magnetic field, one can design a four-level QD with appropriate energy levels, suitable for controlling the optical bistability and multistability by terahertz signal field. The calculated QD structures that we report here can be the base components in photoconductive THz emitters/detectors providing critical fundamental and practical applications at the forefront of scientific knowledge (sensors, flexible electronics, security systems, biomedicine, and others).

Additionally, at the end of the Conclusions section we have added the following sentence:

The QD nanostructures studied here, combined with electric and magnetic field effects, can be the basis for the design of a four-level quantum dot with appropriate energy levels, suitable for controlling the optical bistability and multistability by terahertz signal field. Likewise, these structures find potential applications in developing highly efficient solar cells and can even be the basis for components in photoconductive THz emitters/detectors. We hope that our research will stimulate future theoretical and experimental investigations about electronic, impurity, and excitonic states in conical QDs with potential applications in nanoelectronics.

The Referee:

A few, indicative comments/suggestions can be found below:

  1. What do the authors mean by “eigenvalues differential equations”? I guess they mean that they use the eigenvalues to solve the differential equation system. However, the way it is written both in the abstract and in the conclusions does not make sense to me. Maybe, the authors could consider not using it as a term.

Our reply:

We have modified the manuscript considering the referee's observation. We have stopped using the term "eigenvalues differential equations" and have changed it to "Schrödinger equation"

The Referee:

  1. The manuscript needs to be checked again for inconsistencies and typos. I.e., in line 86 we find: “In Ref. [Review]…”. In lines 137-139 it seems that some notes have been forgotten there.

Our reply:

We thank the Referee for this observation. We apologize for these inadvertent oversights. We have corrected the manuscript. The reference has been correctly cited in the revised version of the manuscript. The Spanish text has been removed.

The Referee:

  1. I understand that this is a theoretical work, and this is fine. However, since the authors study QDs which are highly relevant to applications, they should devote a few lines (maybe in the conclusion and the introduction) to discuss the application relevance and implications of their findings.

Our reply:

We thank the Referee for this observation. At the end of the results and discussion section we have added the following paragraph concerning the possible applications of our quantum dot system:

By size and shape control of the truncated conical QD and external voltage, combined with an externally applied magnetic field, one can design a four-level QD with appropriate energy levels, suitable for controlling the optical bistability and multistability by terahertz signal field. The calculated QD structures that we report here can be the base components in photoconductive THz emitters/detectors providing critical fundamental and practical applications at the forefront of scientific knowledge (sensors, flexible electronics, security systems, biomedicine, and others).

Additionally, at the end of the Conclusions section we have added the following sentence:

The QD nanostructures studied here, combined with electric and magnetic field effects, can be the basis for the design of a four-level quantum dot with appropriate energy levels, suitable for controlling the optical bistability and multistability by terahertz signal field. Likewise, these structures find potential applications in developing highly efficient solar cells and can even be the basis for components in photoconductive THz emitters/detectors. We hope that our research will stimulate future theoretical and experimental investigations about electronic, impurity, and excitonic states in conical QDs with potential applications in nanoelectronics.

The Referee:

  1. In the first model, the authors consider a QD where axial symmetry is preserved, and they consider all the positions of the impurity on this axis. Is this choice only due to mathematical convenience? What are the implications of this symmetry in the behavior of the system? How the results would be modified if the impurity was not placed on the symmetry axis.

Our reply:

The Referee is absolutely right. We have considered a system where axial symmetry is preserved. This has very large implications for the numerical treatment of the problem. In this way, we can use an axis-symmetric model to solve the Schrödinger equations for electron and hole. In this way, the numerical calculation is made only in the middle of the projection in the y = 0 plane of the structure under study, as shown in Figs. 1(a), 1(c) and 1(d). With this, the number of nodes and boundary points in the mesh used in the FEM is of the order of a few tens of thousands to guarantee the convergence of our results. By removing the impurity from the system axis, the axial symmetry is broken, and in this case, the axis-symmetric model no longer works. In that case, then we must proceed to solve the differential equation on the entire volume of the QD, including the surrounding region. We thus pass to a number of nodes and border points that exceeds hundreds of thousands, close to a million, in order to guarantee convergence. This fact in computational times is quite significant even to obtain electron and hole states. Furthermore, it would make it almost impossible, within a reasonable time, to obtain adequate information on excitonic states.

Now, what are the implications of removing the impurity from the axis? Of course, that would be more realistic when compared to experiments. Situations such as the following could be observed: 1) The binding energies of the electron-impurity system decrease due to the repulsive effect of the potential barriers, 2) the quantum number of the ground state of the hole would be modified as a function of the applied magnetic field, 3) there would be changes, although not very appreciable, in the overlap integral and the electron-hole Coulomb energy. In our opinion, there would be novel results, and these could be addressed in future research.

The Referee:

  1. In Fig. 2a, it seems that at R2=R1 where the QD becomes cylindrical, the electron energies for z=0nm and z=15nm cross (reasonable and expected). However, it is not clearly explained why, as R2 increases and R2>R1, the two energies still, approximately, coincide. In (b) the situation is clear. Does this hold for larger R2 values? If this a matter of bad resolution, consider adding an inset zooming at the region 15-to-20nm.

Our reply:

The referee is absolutely right. When R2> R1 it is not possible that the energy of the electron for impurities in the two caps, upper and lower, is the same. This is a scale problem that we have solved by following the Referee's observation and we have included an inset in Fig. 2 (a). In the caption of Fig. 2 we have added the following text:

The inset in Fig. 2(a) shows, with better resolution, that the electron ground state energy for impurities at $z_i=0$ and $z_i=15$\,nm should be different when $R_2>R_1$, as clearly seen in Fig. 2(b) for the case of the hole.

The Referee:

  1. In Fig. 5(a) the “no impurity” and “z=0” seem to begin from the same value. The same holds for the “z=7.5” and “z=15” cases. I’m not sure that the mechanism for this is adequately explained in the text.

Our reply:

The Referee's assessment and his/her comment are absolutely correct. It is not possible that the Coulomb energy between the electron and the hole are exactly the same when $R_2=0$ for the situation "not impurity" and $z_i=0$ and for the situation $z_i=7.5$\,nm and $z_i=15$\,nm.

We want to highlight that for $R_2=0$, the situation with the highest confinement on the hoe-electron pair occurs. There, the conical part of the QD at $z=0$ generates extreme repulsion on the electron and hole wave functions making the effects of an impurity located at $z_i=0$ almost undetectable. This explains the first case of almost equal energies for "without impurity" and $z_i=0$”. When talking about an extreme confinement, the electron and hole wave functions are difficult to deform due to impurity effects when it goes from $z_i=7.5$\,nm to $z_i=15$\,nm., which explains the second case of energies almost equal to close to $20$\,meV.

In consideration of the above, we have decided to add an inset in Fig. 5(a) to show that really in $R_2=0$ the four energies are different and that the problem comes from resolution effects in the Figure. In the caption of Fig. 5 we have added the following text:

"The inset in Fig. 5(a) shows in detail the situation $R_2\sim 0$ where it is evident that the four energies are different despite having two almost equal pairs."

In the text of Fig. 5 we have added the following comment:

We want to highlight that for $R_2=0$, the situation with the highest confinement on the hoe-electron pair occurs. The conical part of the QD at $z=0$ generates extreme repulsion on the electron and hole wave functions making the effects of an impurity located at $z_i=0$ almost undetectable. This fact explains the first case of almost equal energies for "without impurity" and $z_i=0$”. When discussing extreme confinement, the electron and hole wave functions are difficult to deform due to impurity effects when it goes from $z_i=7.5$\,nm to $z_i=15$\,nm. This situation explains the second case of energies almost equal to close to $20$\,meV.

The Referee:

  1. As a final comment, which has to do with the presentation of the manuscript, I would recommend using the same axis font (and style) in all figures. Currently, Figs. 2,3,4 have a quite different format from Figs. 5,6,7,8,9.

Our reply:

We want to thank the Referee for his comment and suggestion. We have changed the format of Figs. 2 and 3.

Reviewer 2 Report

This paper presents a computational study of electron and hole states in conical semiconductor quantum dots. The energies and wave functions of electrons and holes are calculated as a function of quantum dot geometry, applied magnetic field, and position of an impurity, for a GaAs/AlGaAs system. For a CdSe/CdTe system, electron and hole states are calculated as a function of core shell thickness. In addition to the electron and hole states themselves, their Coulomb interaction energy (which approximates the exciton binding energy) is calculated. It is found that the electron and hole distributions can be controlled by the impurity position and quantum dot geometry. In particular, in the CdSe/CdTe system it is possible to achieve direct or indirect excitons by suitable design of the dot. 

All in all, this paper is well written and easy to understand, and the results are intuitively clear. The interplay between quantum confinement, interaction with the impurity, and magnetic field effects is very straightforward. Even though this paper does not really contain much novel physics, it is a nice case study that illustrates how the finite-element method can be useful for describing the electronic properties of quantum dots.

Before I can recommend publication I have a few comments for the authors to consider.

1. Within the effective-mass approximation, the coupling of a magnetic field requires a material-dependent g-factor which can be very different from the g-factor for free electrons (g=2). What values of the g-factors were used in the different semiconductor materials?

2. Strictly speaking, the calculation of excitons in a quantum dot requires the solution of a 2-particle Schroedinger equation (or the diagonalization of a 2-particle Hamiltonian), see for example Z.M. Schultz and J.M. Essick, American Journal of Physics 76, 241 (2008). By contrast, here the exciton wave function is estimated by simply calculating the Coulomb interaction energy of the lowest electron and hole states; this is a very crude approximation, especially for small dots. The authors should discuss this in more detail, and give an estimate of the errors that are made within this approximation.

3. The authors use the term "correlated" and "uncorrelated" in a non-standard way. Correlation is a quantum mechanical effect that occurs in interacting many-body systems, describing Coulomb interaction effects beyond Hartree-Fock exchange. Here, by contrast, the authors use "correlation" to refer to electrons and holes in the presence of a fixed impurity potential (see lines 275, 279, and 472), or for interacting electrons and holes (line 463). In all those cases "correlation" should probably be replaced with "interaction".

4. Some minor points:
* line 32: "Another effect which several authors have been interested in corresponds to the presence of shallow donor and...."
* line 86: a reference is missing.
* lines 137-139: this should probably not be there.
* line 192: subsections 3.1, 3.2 instead of A and B
* line 287: it is not clear what is meant by "oscillatory character". Perhaps this can be rephrased.

Author Response

Referee 2:

The Referee:

This paper presents a computational study of electron and hole states in conical semiconductor quantum dots. The energies and wave functions of electrons and holes are calculated as a function of quantum dot geometry, applied magnetic field, and position of an impurity, for a GaAs/AlGaAs system. For a CdSe/CdTe system, electron and hole states are calculated as a function of core shell thickness. In addition to the electron and hole states themselves, their Coulomb interaction energy (which approximates the exciton binding energy) is calculated. It is found that the electron and hole distributions can be controlled by the impurity position and quantum dot geometry. In particular, in the CdSe/CdTe system it is possible to achieve direct or indirect excitons by suitable design of the dot.

All in all, this paper is well written and easy to understand, and the results are intuitively clear. The interplay between quantum confinement, interaction with the impurity, and magnetic field effects is very straightforward. Even though this paper does not really contain much novel physics, it is a nice case study that illustrates how the finite-element method can be useful for describing the electronic properties of quantum dots.

Before I can recommend publication I have a few comments for the authors to consider.

Our reply:

We are deeply grateful to the referee as we consider that he/she has made a very careful reading of our manuscript and has presented us with very valuable suggestions. By considering the Referee's suggestions, we believe that our manuscript has been improved in a very important way. We hope that the new version of our article will be seen by the Referee as suitable for publication in the journal Nanomaterials.

The Referee:

  1. Within the effective-mass approximation, the coupling of a magnetic field requires a material-dependent g-factor which can be very different from the g-factor for free electrons (g=2). What values of the g-factors were used in the different semiconductor materials?

Our reply:

We thank the Referee for his comment. In the present study, and despite considering magnetic field effects, we have omitted the Zeeman effect on the system. Consequently, we have not used the g-factor values that correspond to the study materials and are different from the value in bulk given the confinement associated with the heterostructure. In the paragraph following Eq. (2) of the revised version of the manuscript, we have added the following text with its corresponding References:

In previous studies reported in the literature, we have analyzed the hydrostatic pressure and size effects on the conduction-electron g-factor in GaAs-Ga$_{1-x}$Al$_x$As quantum wells under magnetic field. Clearly, the results have shown that given the quantum confinement, the $g$-factor changes drastically with the size of the structure concerning its value in bulk and that hydrostatic pressure is an excellent tool by which the $g$-factor magnitude, and even the sign, can be manipulated \cite{gfactor_1,gfactor_2}. In this article, we focus our interest on the effects of impurities and their position, the shape and size of the QDs, as well as the type of coupling between the materials that make up the heterostructure (type-I and type-II heterostructures) for electrons, holes, and excitons confined in conical QDs. We have omitted the Zeeman effect despite considering magnetic field effects, which gives rise to a fine splitting of the energy level structure.

\bibitem{gfactor_1}\textcolor{red}{Porras-Montenegro, N.; Duque, C.A.;  Reyes-G\'omez, E.; Oliveira, L.E. Effects of hydrostatic pressure on the electron $g_{\|}$ factor and $g$-factor anisotropy in GaAs-(Ga,Al)As quantum wells under magnetic fields. {\em J. Phys.: Condens. Matter} {\bf 2008}, {\em 20}, 465220.}

\bibitem{gfactor_2}\textcolor{red}{Porras-Montenegro, N.; Raigoza, N.; Reyes-G\'omez, E.; Duque, C.A.; Oliveira, L.E. Effects of hydrostatic pressure on the conduction-electron $g$-factor in GaAs-Ga$_{1-x}$Al$x$As quantum wells. {\em Phys. Status Solidi B} {\bf 2009}, {\em 246}, 648--651.}

The Referee:

  1. Strictly speaking, the calculation of excitons in a quantum dot requires the solution of a 2-particle Schroedinger equation (or the diagonalization of a 2-particle Hamiltonian), see for example Z.M. Schultz and J.M. Essick, American Journal of Physics 76, 241 (2008). By contrast, here the exciton wave function is estimated by simply calculating the Coulomb interaction energy of the lowest electron and hole states; this is a very crude approximation, especially for small dots. The authors should discuss this in more detail, and give an estimate of the errors that are made within this approximation.

Our reply:

We thank the Referee for his/her comment. We fully agree with the Referee that a perturbative calculation of the Coulomb interaction between the electron and the hole confined in the structure is very crude, possibly too simplified considering the immense amount of works reported in the literature where the authors report developments quite complete. In the revised version of the manuscript, in the paragraph after Eq. (4), we have added the following text with the corresponding references:

We want to say that in this article we have followed the same strategy previously used in conical QDs subjected to combined effects of electric and magnetic fields \cite{Heyn_2}. In Ref. \cite{Heyn_2}, the calculations of the electron and hole states using the FEM were contrasted with results obtained through a diagonalization process with an orthonormal basis composed of a product between Bessel and sinusoidal functions. With a coincidence of up to $0.1$\,meV. Additionally, comparisons were made with a previous more elaborated model \cite{Graf2014}, which considers 8 bands $\vec{k}\cdot\vec{p}$ theory and configuration interaction with deviations only by a few percent. In Ref. \cite{Heyn_2} and in this work, the electron-hole Coulomb interaction obtained via a perturbative calculation was corroborated with variational calculations considering two types of trial functions: \textit{i}) a hydrogenic-like function with one variational parameter \cite{Duque1, Duque2} and \textit{ii}) a trial function constructed by the product of two independent Gaussian functions. In this second case, two variational parameters were used to describe the radial problem and the problem along the axial direction \cite{Duque3, Duque4} separately. The results obtained by the perturbative calculation coincided with the variational ones in 97\% for the binding energy.

\bibitem{Graf2014}\textcolor{red}{Graf, A.; Sonnenberg, D.; Paulava, V.; Schliwa, A.; Heyn, Ch.; Hansen, W. Excitonic states in GaAs quantum dots fabricated by local droplet etching. {\em Phys. Rev. B} {\bf 2014}, {\em 89}, 115314.}

\bibitem{Duque1}\textcolor{red}{L\'{o}pez, S.Y.; Porras-Montenegro, N.; Duque, C.A. Excitons in coupled quantum dots: hydrostatic pressure and electric field effects. \textit{Phys. Status Solidi B} \textbf{2009}, \textit{246}, 630--634.}

\bibitem{Duque2}\textcolor{red}{L\'{o}pez, S.Y.; Mora-Ramos, M.E.; Duque, C.A. Photoluminescence energy transitions in GaAs-Ga$_{1-x}$Al$_x$As double quantum wells: Electric and magnetic fields and hydrostatic pressure effects. \textit{Physica B} 2009, \textbf{404}, 5181--5184.}

\bibitem{Duque3}\textcolor{red}{Mora-Ramos, M.E.; Barseghyan, M.G.; Duque, C.A. Excitons in cylindrical GaAs P\"{o}schl-Teller quantum dots: hydrostatic pressure and temperature effects. \textit{ Physica E} \textbf{2010} \textit{43}, 338--344.}

\bibitem{Duque4}\textcolor{red}{Mora-Ramos, M.E.; Barseghyan, M.G; Duque, C.A. Excitons in a cylindrical GaAs P\"{o}schl-Teller quantum dot.  \textit{Phys. Status Solidi B} \textbf{2011} \textit{248}, 1412–-1419.}

The Referee:

  1. The authors use the term "correlated" and "uncorrelated" in a non-standard way. Correlation is a quantum mechanical effect that occurs in interacting many-body systems, describing Coulomb interaction effects beyond Hartree-Fock exchange. Here, by contrast, the authors use "correlation" to refer to electrons and holes in the presence of a fixed impurity potential (see lines 275, 279, and 472), or for interacting electrons and holes (line 463). In all those cases "correlation" should probably be replaced with "interaction".

Our reply:

We have followed the recommendation of the Referee and therefore we have made the corresponding adjustments in the revised version of the manuscript.

The Referee:

  1. Some minor points:
    * line 32: "Another effect which several authors have been interested in corresponds to the presence of shallow donor and...."
    * line 86: a reference is missing.
    * lines 137-139: this should probably not be there.
    * line 192: subsections 3.1, 3.2 instead of A and B

* line 287: it is not clear what is meant by "oscillatory character". Perhaps this can be rephrased.

Our reply:

We want to thank the Referee for the highly careful review that he/she has done to our manuscript and that has helped us to correct mistakes that we have truly made due to carelessness and inattention. All the corrections recommended by the referee in the item 4 have been implemented in the revised version of the manuscript.

Reviewer 3 Report

                                      Referre report

Manuscript ID: nanomaterials-1404739
Title: Shallow-donor impurity states with excitonic contribution in 
GaAs/AlGaAs and CdTe/CdSe truncated conical quantum dots under applied magnetic field
Authors: Lorenz Pulgar-Velasquez, Jose Sierra-Ortega, J. A. Vinasco, D. Laroze, A. Radu, E. Kasapoglu, R. L. Restrepo, John A. Gil Corrales, A. L. Morales, C. A. Duque

The manuscript is devoted to analyzing the influence of truncated conical geometry on the electronic states in the conduction and valence bands.

Two structures are considered

  1. GaAs surrounded by AlGaAs.
  2. CdSe/CdTe core-shell quantum dots.

In both cases, the influence of an external magnetic field along the grown direction on the electron and the heavy-holes states is considered.

Within the framework of the envelope function approximation, the finite element method was implemented to solve the corresponding Schrödinger equation. For the structure of GaAs / AlGaAs, the presence of shallow-impurity and the excitonic effect are evaluated. The excitonic effect is treated in the first order perturbation theory.

The dependence of the energy on the lower radius of the structure for the electron and the hole ground state in the presence of shallow-impurity is analyzed. Also, the effects of geometric shape and impurity on the heavy-hole exciton are clearly shown. The work is extended to CdSe/CdTe core-shell quantum dots.

It is shown graphically how the electronic states change in a truncated canonical form of GaAs / AlGaAs as a function of an applied external magnetic field.

Undoubtedly, the numerical effort for the evaluation of the electronic states including an impurity (wave functions and eigen-energies) as a function of the complex geometry is significant. This is one of the relevant results of the present manuscript.

I have the following remarks.

1.- In the Abstract the authors states.

 “This study shows that the magnetic field and donor impurities are relevant factors in the optoelectronic properties of conical quantum dots. ”

Unfortunately, this claim is not disputed in the manuscript. The influence of impurities on electronic states, exciton energy, etc. are studied, but the implication of these results on optoelectronic properties is not discussed. I recommend to add a few sentences that show the connections between the results obtained and the field of optoelectronics.

2.- The ground state of the heavy hole exciton is evaluated using perturbation theory. Based on the solutions of the Schrödinger equation including an impurity that interact with electrons or holes, the authors evaluate the Coulomb interaction using the equation Eq. (3). It is important to remember that the perturbation theory in this case is not completely justified. Comparing Figs. (2) and (3) for the energy of electrons and holes with the result of Fig. 5 it can be seen that the Coulomb energy of electrons and holes has the same order of magnitude as the binding energy of electrons and holes. Depending on the distance, R_2, and the impurity position, Eqs. (3) or (4) cannot be applied. The same can be argued, when an external magnetic field is applied.

  1. There are a number of terminological and typos to be corrected throughout the whole manuscript. Authors must consider the following corrections / changes.

- In line 6 on p. 3 is missing the reference “….In Ref. [Review]…..”

-On p.5

  1. i) In line 19 the constant c=w/b is defined but it is not mentioned in the manuscript.
  2. ii) The paragraph is written in Spanish “Tenemos que anotar que en el caso de los core-shell QDs formados de CdTe y CdSe, en el caso de los electrones la región del pozo corresponde a ”.

iii) Below Eq. (2), “where l ∈ Z is the principal quantum number...” it should be, l is the azimuthal quantum number.

In conclusion, the manuscript cannot be recommended for publication in

the present form. It can be considered after a revision along the

above remarks.

Author Response

Referee 3

The Referee:

The manuscript is devoted to analyzing the influence of truncated conical geometry on the electronic states in the conduction and valence bands.

Two structures are considered

  1. GaAs surrounded by AlGaAs.
  2. CdSe/CdTe core-shell quantum dots.

In both cases, the influence of an external magnetic field along the grown direction on the electron and the heavy-holes states is considered.

Within the framework of the envelope function approximation, the finite element method was implemented to solve the corresponding Schrödinger equation. For the structure of GaAs / AlGaAs, the presence of shallow-impurity and the excitonic effect are evaluated. The excitonic effect is treated in the first order perturbation theory.

The dependence of the energy on the lower radius of the structure for the electron and the hole ground state in the presence of shallow-impurity is analyzed. Also, the effects of geometric shape and impurity on the heavy-hole exciton are clearly shown. The work is extended to CdSe/CdTe core-shell quantum dots.

It is shown graphically how the electronic states change in a truncated canonical form of GaAs / AlGaAs as a function of an applied external magnetic field.

Undoubtedly, the numerical effort for the evaluation of the electronic states including an impurity (wave functions and eigen-energies) as a function of the complex geometry is significant. This is one of the relevant results of the present manuscript.

I have the following remarks.

Our reply:

We are deeply grateful to the referee as we consider that he/she has made a very careful reading of our manuscript and has presented us with very valuable suggestions. By considering the Referee's suggestions, we believe that our manuscript has been improved in a very important way. We hope that the new version of our article will be seen by the Referee as suitable for publication in the journal Nanomaterials.

The Referee:

1.- In the Abstract the authors states.

“This study shows that the magnetic field and donor impurities are relevant factors in the optoelectronic properties of conical quantum dots.”

Unfortunately, this claim is not disputed in the manuscript. The influence of impurities on electronic states, exciton energy, etc. are studied, but the implication of these results on optoelectronic properties is not discussed. I recommend to add a few sentences that show the connections between the results obtained and the field of optoelectronics.

Our reply:

At the end of the results and discussion section we have added the following paragraph concerning the possible applications of our quantum dot system:

By size and shape control of the truncated conical QD and external voltage, combined with an externally applied magnetic field, one can design a four-level QD with appropriate energy levels, suitable for controlling the optical bistability and multistability by terahertz signal field. The calculated QD structures that we report here can be the base components in photoconductive THz emitters/detectors providing critical fundamental and practical applications at the forefront of scientific knowledge (sensors, flexible electronics, security systems, biomedicine, and others).

At the end of the Conclusions section we have added the following sentence:

The QD nanostructures studied here, combined with electric and magnetic field effects, can be the basis for the design of a four-level quantum dot with appropriate energy levels, suitable for controlling the optical bistability and multistability by terahertz signal field. Likewise, these structures find potential applications in developing highly efficient solar cells and can even be the basis for components in photoconductive THz emitters/detectors. We hope that our research will stimulate future theoretical and experimental investigations about electronic, impurity, and excitonic states in conical QDs with potential applications in nanoelectronics.

The Referee:

2.- The ground state of the heavy hole exciton is evaluated using perturbation theory. Based on the solutions of the Schrödinger equation including an impurity that interact with electrons or holes, the authors evaluate the Coulomb interaction using the equation Eq. (3). It is important to remember that the perturbation theory in this case is not completely justified. Comparing Figs. (2) and (3) for the energy of electrons and holes with the result of Fig. 5 it can be seen that the Coulomb energy of electrons and holes has the same order of magnitude as the binding energy of electrons and holes. Depending on the distance, R_2, and the impurity position, Eqs. (3) or (4) cannot be applied. The same can be argued, when an external magnetic field is applied.

Our reply:

The authors want to thank the Referee for this very important observation which has to do with the validity regime of the calculations that we report here. The strongest approach that we have made, in our opinion, and according to the Referee's point of view, is to develop a first-order perturbative calculus of the Coulomb interaction between the electron and the hole. We fully agree with the Referee that a perturbative calculation of the Coulomb interaction between the electron and the hole confined in the structure is very crude, possibly too simplified considering the immense amount of works reported in the literature where the authors report developments quite complete.

In consideration of this, at the end of section 3.2 we have decided to add the following paragraph.

Finally, we want to emphasize that in those situations where the independent particle energies for the electron and the hole are of the same order of magnitude as the Coulomb interaction between both charge carriers, it is clear that the results obtained by means of a first order perturbative calculation should be viewed with some reserve and only as a guiding information to analyze the physics of the problem in question.

Additionally, in the revised version of the manuscript, in the paragraph after Eq. (4), we have added the following text with the corresponding references:

We want to say that in this article we have followed the same strategy previously used in conical QDs subjected to combined effects of electric and magnetic fields \cite{Heyn_2}. In Ref. \cite{Heyn_2}, the calculations of the electron and hole states using the FEM were contrasted with results obtained through a diagonalization process with an orthonormal basis composed of a product between Bessel and sinusoidal functions. With a coincidence of up to $0.1$\,meV. Additionally, comparisons were made with a previous more elaborated model \cite{Graf2014}, which considers 8 bands $\vec{k}\cdot\vec{p}$ theory and configuration interaction with deviations only by a few percent. In Ref. \cite{Heyn_2} and in this work, the electron-hole Coulomb interaction obtained via a perturbative calculation was corroborated with variational calculations considering two types of trial functions: \textit{i}) a hydrogenic-like function with one variational parameter \cite{Duque1, Duque2} and \textit{ii}) a trial function constructed by the product of two independent Gaussian functions. In this second case, two variational parameters were used to describe the radial problem and the problem along the axial direction \cite{Duque3, Duque4} separately. The results obtained by the perturbative calculation coincided with the variational ones in 97\% for the binding energy.

\bibitem{Graf2014}\textcolor{red}{Graf, A.; Sonnenberg, D.; Paulava, V.; Schliwa, A.; Heyn, Ch.; Hansen, W. Excitonic states in GaAs quantum dots fabricated by local droplet etching. {\em Phys. Rev. B} {\bf 2014}, {\em 89}, 115314.}

\bibitem{Duque1}\textcolor{red}{L\'{o}pez, S.Y.; Porras-Montenegro, N.; Duque, C.A. Excitons in coupled quantum dots: hydrostatic pressure and electric field effects. \textit{Phys. Status Solidi B} \textbf{2009}, \textit{246}, 630--634.}

\bibitem{Duque2}\textcolor{red}{L\'{o}pez, S.Y.; Mora-Ramos, M.E.; Duque, C.A. Photoluminescence energy transitions in GaAs-Ga$_{1-x}$Al$_x$As double quantum wells: Electric and magnetic fields and hydrostatic pressure effects. \textit{Physica B} 2009, \textbf{404}, 5181--5184.}

\bibitem{Duque3}\textcolor{red}{Mora-Ramos, M.E.; Barseghyan, M.G.; Duque, C.A. Excitons in cylindrical GaAs P\"{o}schl-Teller quantum dots: hydrostatic pressure and temperature effects. \textit{ Physica E} \textbf{2010} \textit{43}, 338--344.}

\bibitem{Duque4}\textcolor{red}{Mora-Ramos, M.E.; Barseghyan, M.G; Duque, C.A. Excitons in a cylindrical GaAs P\"{o}schl-Teller quantum dot.  \textit{Phys. Status Solidi B} \textbf{2011} \textit{248}, 1412–-1419.}

The Referee:

3.- There are a number of terminological and typos to be corrected throughout the whole manuscript. Authors must consider the following corrections / changes.

- In line 6 on p. 3 is missing the reference “….In Ref. [Review]…..”

-On p.5

  1. i) In line 19 the constant c=w/b is defined but it is not mentioned in the manuscript.
  2. ii) The paragraph is written in Spanish “Tenemos que anotar que en el caso de los core-shell QDs formados de CdTe y CdSe, en el caso de los electrones la región del pozo corresponde a ”.
  3. iii) Below Eq. (2), “where l ∈ Z is the principal quantum number...” it should be, is the azimuthal quantum number.

Our reply:

  1. In Eq. (1), the super-index c for the effective mass indicates the dot material effective mass (c = w) or the barrier material effective mass (c = b). At the beginning of section 3.1 we referred to the effective masses in the dot and barrier regions.
  2. We regret that we missed a text in Spanish in the manuscript. This is the result of an oversight that we must avoid in the future. We apologize to the Referee and thank you for pointing out this error to us.
  3. The main principal quantum number has been changed by azimuthal quantum number

The Referee:

In conclusion, the manuscript cannot be recommended for publication in the present form. It can be considered after a revision along the above remarks.

Our reply:

We hope that the new revised version of our manuscript will be seen by the Referee as suitable for publication in the journal Nanomaterials.

Round 2

Reviewer 1 Report

In the revised version the authors provided sufficient answers to my comments and I recommend the publication to Nanomaterials.

Reviewer 3 Report

The manuscript has been sufficiently improved to warrant publication in Nanomaterials